# The Role of Vitamin D in Early Knee Osteoarthritis and Its Relationship with Their Physical and Psychological Status

**DOI:** 10.3390/nu13114035

**Published:** 2021-11-12

**Authors:** Ana Alabajos-Cea, Luz Herrero-Manley, Luis Suso-Martí, Enrique Viosca-Herrero, Ferran Cuenca-Martínez, Clovis Varangot-Reille, María Blanco-Díaz, Joaquín Calatayud, José Casaña

**Affiliations:** 1Servicio de Medicina Física y Rehabilitación, Hospital La Fe, 46026 Valencia, Spain; alabajos.ana@gmail.com (A.A.-C.); luzherreromanley@gmail.com (L.H.-M.); eviosca@gmail.com (E.V.-H.); 2Grupo de Investigación en Medicina Física y Rehabilitación, Instituto de Investigación Sanitaria La Fe (IISLAFE), 46026 Valencia, Spain; 3Exercise Intervention for Health Research Group (EXINH-RG), Department of Physiotherapy, University of Valencia, 46026 Valencia, Spain; fecuen2@gmail.com (F.C.-M.); clovis.varangotreille@gmail.com (C.V.-R.); joaquin.calatayud@uv.es (J.C.); jose.casana@uv.es (J.C.); 4Surgery and Medical Surgical Specialities Department, Faculty of Medicine and Health Sciences, University of Oviedo, 33006 Oviedo, Spain; blancomaria@uniovi.es

**Keywords:** osteoarthritis, early osteoarthritis, vitamin D

## Abstract

Osteoarthritis (OA) is a common joint condition and one of the greatest causes of disability worldwide. The role of vitamin D in the origin and development of the disease is not clear, although it could have important implications for diagnosis and treatment. For this proposal, a cross-sectional study with a non-probabilistic sample was performed. In total, 48 with early osteoarthritis (EOA) and 48 matched controls were selected, and serum 25(OH)D and parathyroid hormone (PTH) levels were analyzed. In addition, physical and psychological variables were measured to establish their relationship with vitamin D levels. Patients with EOA showed lower levels (22.3 ± 7.3 ng/mL) in comparison to matched controls (29.31 ± 9.2 ng/mL). A statistically significant higher number (Chi-squared = 8.525; *p* = 0.004) of patients with EOA had deficiency levels (<20 ng/mL) compared to the control group. Patients with lower vitamin D levels showed higher levels of pain intensity, disability, and anxiety, as well as poorer values for sit-to-stand, walking speed, and social participation. Correlation analysis showed a relationship between serum 25(OH)D, PTH and pain intensity, and social participation. These results highlight the relevance of vitamin D in the early diagnosis and prevention of EOA.

## 1. Introduction

Osteoarthritis (OA) is defined as a common joint condition that typically causes chronic pain and is one of the greatest causes of disability worldwide, being considered a major public health challenge [1]. Knee OA has been associated with degenerative changes due to the progressive loss of articular cartilage, subchondral bone changes, synovial inflammation, and meniscus degeneration [2,3]. It is of prime importance to find predictors for knee OA progression to improve treatment options and to prevent this disabling disease [4,5].

Unfortunately, although some theories regarding the pathogenesis of OA do exist, the exact mechanism is not well charted [6]. The most frequent nutritional deficiency worldwide is vitamin D deficiency (defined as serum level of 25-hydroxyvitamin D [25-(OH) D] < 20 nmol/L) [7]. Specifically, more than 40% of the adult population over the age of 50 has this deficiency, being lower in women [8]. It is well known that vitamin D has an important influence on the state of many articular structures, such as cartilage and subchondral bone, as well as muscle tissues, all of which play a part in the progression of knee OA [9]. Similarly, parathyroid hormone levels (PTH) could play an important role in the loss and formation of bone and these levels have been related to subchondral bone remodeling, and therefore, to the radiological and clinical progression of knee OA [10].

In relation to this, previous studies suggest that vitamin D could stimulate proteoglycan synthesis in mature chondrocytes through a metabolic transformation via vitamin D receptors [11,12]. Moreover, vitamin D deficiency could impede bone remodeling causing OA through complex pathophysiological processes, and other studies suggested that low serum vitamin D could be associated with an increased risk of knee OA progression [13]. Conversely, vitamin D supplementation could be able to prevent the progression of OA by increasing bone remodeling and also by reducing the abnormal pathophysiological process [14]. Additionally, it was suggested that supplementation may help improve disease progression in patients with vitamin D deficiency but does not have any additional benefits on those with optimal levels of vitamin D [15]. However, recent studies showed that low levels of vitamin D are not associated with the risks of developing knee OA, except perhaps with the progression of knee OA [12,16]. In addition, vitamin D supplementation may not help to control pain or to reduce structural disease progression in this condition [16,17].

It has been suggested that the structural findings in OA are a late phenomenon [18,19]. For this reason, non-surgical treatments for OA are often ineffective, which may be due in part to its use late in the course of the disease, when structural deterioration is usually advanced [18]. Therefore, there is a great scientific and social interest in ‘early osteoarthritis’ (EOA), as it is believed that identifying patients affected by EOA to initiate early interventions and therapeutic approaches could prevent progression and severe structural changes in the joint associated with later stages of OA [20].

Most of the studies that have reported a link between vitamin D and OA have been conducted in patients with advanced stages of the disease; however, little is understood regarding the effect that vitamin D and PTH could have on the initiation and progression of EOA. Early identification of these factors could be critical for early diagnosis, which would facilitate early treatment to prevent the dramatic development of knee OA. Therefore, the aim of this cross-sectional study was to study the association of serum concentrations of vitamin D with knee EOA. The secondary objective was to determine the association of serum concentrations of PTH with knee EOA and to determine the relationship between vitamin D deficiency, PTH level, and pain intensity, disability, psychological, and functional variables in patients with knee EOA.

## 2. Materials and Methods

### 2.1. Study Design

A cross-sectional study with a non-probabilistic sample was designed. The international recommendations for Strengthening the Reporting of Observational Studies in Epidemiology were followed [20]. The research procedures were performed according to the ethical standards of the Declaration of Helsinki and approved by an Ethics Committee (CEIm La Fe 2017/0147). All participants received an explanation and signed a written informed consent.

### 2.2. Participants

Subjects were recruited at Hospital La Fe, Valencia, Spain, within the H2020 project OACTIVE. The design of the data collection protocol started in November 2017 and lasted until July 2018.

The subjects recruited were evaluated by a group of three experienced physical medicine and rehabilitation clinicians. The inclusion criteria for EOA patients were based on Luyten’s proposal for EOA classification, due to the criteria used found a specificity of 76.5% for detection of clinical progression, being valid criteria for research use [21]. Criteria were as follows: (a) Patient-based questionnaires: knee Injury and osteoarthritis outcome score: 2 out of the 4 KOOS subscales (pain, symptoms, function, or knee-related quality of life) need to score “positive” (≤85%); (b) patients should present joint line tenderness or crepitus in the clinical examination; (c) X-rays: Kellgren and Lawrence (KL) grade 0–1 standing, weight-bearing (at least 2 projections: PA fixed flexion and skyline for patellofemoral OA) [22]. To match with EOA patients, controls healthy subjects with similar descriptive and sociodemographic characteristics were selected. The inclusion criteria were (a) patient age greater than or equal to 40 years; (b) KL grade 0–1.

Exclusion criteria were: (a) Any cognitive disability that hindered viewing of the audio-visual material; (b) illiteracy; (c) comprehension or communication difficulties, (d) insufficient Spanish language comprehension to follow measurement instructions; (e) presence of any rheumatic, autoimmune or infectious pathology.

### 2.3. Outcome Measures

#### 2.3.1. Vitamin D and Parathyroid Hormone Levels

For the collection of blood, separation of plasma, and long storage of the samples, we followed the rules proposed by the Standard Operating Procedures Internal Working Group (SOPIWG)/Early Detection Research Network (EDRN) for specimen collection. Four aliquots were sent on dry ice by courier to the Biobanco La Fe and kept at −50 °C until measurements of serum intact PTH and 25OHD were taken using commercial ELISA kits (IDS Co., Bolden, UK), with intraassay coefficients of variation ranging from 5% to 15%.

Serum 25(OH)D and PTH concentrations were determined by radioimmunoassay. Vitamin D levels were classified as follows: vitamin D deficiency, serum 25(OH)D concentration < 20 ng/mL; vitamin D insufficiency, serum 25(OH)D concentration 20–29 ng/mL; and vitamin D sufficiency, serum 25(OH)D concentration ≥ 30 ng/mL [23,24].

#### 2.3.2. Pain and Disability Variables

##### Pain Intensity

To measure pain intensity, the Visual Analogue Scale (VAS) was used. The scale is a 100-mm line with two endpoints representing the extreme states “no pain” (0) and “the maximal pain imaginable” (10). The VAS scale has been shown to have good retest reliability (r = 0.94, *p* < 0.001) [25,26].

##### Western Ontario and McMaster Universities Osteoarthritis Index (WOMAC)

The WOMAC questionnaire is self-administered and is used to assess OA disability. It is composed of 24 items divided into 3 aspects: functional pain (5 items), stiffness (2 items), and activities of daily life difficulties (17 items). Higher values mean poorer WOMAC subscales scores of pain and physical function. The Spanish version of the WOMAC questionnaire has an internal consistency of 0.82 for pain and 0.93 for physical function subscales [27].

#### 2.3.3. Functional Variables

##### Five-Time Sit to Stand

To assess functional capacity, the sit-to-stand test was employed. The test was performed as follows: with the patient seated with their back against the back of the chair, the clinician counted each stand aloud so that the patient remained oriented. The clinician stopped the test when the patient achieved the standing position on the 5th repetition [28]. The amount of time needed to realize the test was extracted in seconds.

##### Walking Speed

The subject walked without assistance 10 m (32.8 feet) and the time was measured for the intermediate 6 m (19.7 feet). Assistive devices could be used but had to be kept consistent and documented from test to test. It was performed at the fastest speed possible. There were three trials collected and the average of the three trials was calculated for the measurement [29,30].

#### 2.3.4. Psychological Variables

##### Anxiety and Depression Symptoms

To assess anxiety and depression symptoms, the Hospital Anxiety and Depression Scale (HADS) was used. HADS includes 14 items in a 4-point Likert-type scale. For the full scale, the internal consistency is 0.90. For the depression subscale, the internal consistency is 0.84, and 0.85 for the anxiety subscale [31].

##### Social Participation

To determine the frequency and diversity of social involvement, the Maastricht Social Participation Profile was used [32]. The scale includes 9 items: ‘How often in the past four weeks have you (taken part in/been to): (1) a club, interest group or activity group, church or other similar activity; (2) a cultural or educational event such as the cinema, theatre, museum, talk or course; (3) eaten out; (4) out to a pub, cafe or tearoom; (5) a public event; (6) an organized games afternoon or evening; (7) a day trip organized by a club or society; (8) committee work for a club, society or another group; (9) any organized voluntary work?’ The response categories range from zero (‘not at all’) to three (‘more than twice a week’). The psychometric properties were appropriate (Cronbach’s αbaseline = 0.64; αfollow-up = 0.64), and higher scores equated to higher social participation [32].

### 2.4. Procedures

The procedure and an informed consent form were explained and given to all the participants. The included subjects filled out the sociodemographic questionnaire and self-reported measures of disability, pain, and psychological variables. Finally, the serum 25(OH)D concentrations and functional tests were assessed. This procedure was identical for both groups.

### 2.5. Statistical Analysis

The sociodemographic data of the participants were analyzed. The data were summarized using frequency counts, descriptive statistics, and summary tables. Statistics Package for Social Sciences (SPSS 24, IBM Inc., Armonk, NY, USA) was used for data analysis. The categorical variables are shown as frequencies and percentages. The quantitative results are represented by descriptive statistics (confidence interval, mean, and standard deviation). Student’s *t*-test was used for the group comparisons. Cohen’s d effect sizes were calculated for multiple comparisons of the outcome variables. According to Cohen’s method, the magnitude of the effect was classified as small (0.20–0.49), medium (0.50–0.79), or large (0.80).

The relationships between vitamin D and PTH measures with psychological and physical measurements were analyzed using Pearson’s correlation coefficients. A Pearson’s correlation coefficient greater than 0.60 indicated a strong correlation, a coefficient between 0.30 and 0.60 indicated a moderate correlation, and a coefficient below 0.30 indicated a low or very low correlation [33].

## 3. Results

A total of 96 participants were included in the study, with a mean age of 51.81 ± 5.59 (39 men and 58 women). In total, 48 of the participants met the criteria to be classified as EOA and 48 participants as matched controls. No abandonment of any study participant was recorded nor were adverse effects reported during the assessments. There were no statistically significant differences between the groups in terms of descriptive and demographic variables (Table 1).

### 3.1. Vitamin D and PTH Levels

Regarding serum 25(OH)D concentrations, patients with EOA showed lower levels (22.3 ± 7.3 ng/mL) in comparison to matched controls (29.31 ± 9.2 ng/mL), showing statistically significant differences between groups.

Student’s *t*-test showed statistically significant differences (*p* = 0.03), with a medium effect size (MD: −7.01; d = 0.72). However, no statistically significant differences were found for PTH levels between both groups (MD: −1.42, *p* > 0.05) (Table 2).

Participants in both groups were stratified according to baseline levels based on serum 25(OH)D concentration. A statistically significant higher number (Chi-squared = 8.525; *p* = 0.004) of patients with EOA had deficiency levels compared to the control group. For patients with EOA, 54.2% had values below 20 ng/mL (deficiency), 31.3% had values below 30 ng/mL (insufficiency), and only 14.5% had normal values (<30 mg/mL). Regarding the matched controls, the highest proportion (45.8%) presented values of insufficiency (<30 ng/mL) and normality (33.3%). Only 20.9% presented deficiency in this value (<20 ng/mL) (Figure 1).

### 3.2. Physical and Psychological Status

The 48 patients with EOA were divided according to their vitamin D levels. In total, 26 participants presented deficiency in their levels (<20 ng/mL) and 22 presented higher values of 20 ng/mL.

About pain intensity levels, the *t*-test showed statistically significant differences, showing higher levels of pain intensity in patients with lower vitamin D levels (MD = 2.73; *p* < 0.01; d = 0.88). Similarly, statistically significant differences were found in the WOMAC values with a higher score in the deficient group (MD = 5.51; *p* = 0.032; d = 0.41) (Table 3).

Regarding functional tests, the *t*-test showed statistically significant differences for sit-to-stand and walking speed, showing poorer values for patients with vitamin D deficiency (MD = 5.76; *p* < 0.01; d = 1.90 and MD = 1.92; *p* < 0.01; d = 1.83, respectively) (Table 3).

Finally, regarding psychological variables, patients with vitamin D deficiency showed statistically significant differences for anxiety and depression in comparison to patients with higher vitamin D levels (MD = 3.35; *p* =0.028; d = 0.88 and MD = 2.9; *p* = 0.038; d = 0.94, respectively). In relation to social participation, statistically significant differences were also found, with lower levels for patients with vitamin D deficiency (MD = −4.16; *p* < 0.01; d = 1.01) (Table 3).

The physical and psychological status according to vitamin D levels is summarized in Figure 2.

### 3.3. Correlation Analysis

The correlation analysis was performed in patients with EOA, showing strong correlations between vitamin D levels and pain intensity (r = −0.651; *p* < 0.01) and between vitamin D levels and anxiety (r = −0.623; *p* < 0.01). In addition, moderate correlations were found between vitamin D levels and disability (r = −0.443; *p* < 0.05), depression (r = −0.351; *p* < 0.05), and social participation (r = 0.355; *p* < 0.05). Regarding PTH levels, the strongest correlation was found with pain intensity (r = −0.686; *p* < 0.01). Additionally, a moderate correlation between PTH and social participation was found (r = 0.331; *p* < 0.05) (Table 4).

## 4. Discussion

The primary aim of this cross-sectional study was to compare the level of vitamin D of patients with EOA with matched controls. The secondary objectives were to compare the level of PTH of patients with EOA with matched controls and to determine the relationship between vitamin D deficiency, PTH level, and pain intensity, disability, psychological, and functional variables in patients with knee EOA. Results showed that EOA patients showed statistically significant lower levels of vitamin D than matched controls, but not PTH levels. It was also found that vitamin D deficiency had a statistically significant relationship with a higher level of pain intensity, disability, anxiety, and depressive symptoms, and lower social participation and physical performance. A lower PTH level is associated with higher pain intensity and lower social participation.

Patients with EOA had vitamin D insufficiency with a mean of 22.3 ng/mL, with a majority of patients having deficiency (54.2%). Similar to these results, Wu et al. found in their meta-analysis that patients with chronic pain, including OA, had a lower average vitamin D concentration than controls [21]. They found that patients with OA had a 20% difference in concentration [21]. Patients with a deficiency of vitamin D showed a statistically significant negative strong relationship between vitamin D concentration and pain intensity. The relationship between pain intensity and vitamin D is controversial [22,23,24]. For example, Cakar et al. found no relationship between vitamin D concentration and pain intensity. However, they included patients with KL grade 2 or more [22]. Contrary to them, our study included patients with EOA, so, a deficiency in vitamin D might have greater implication in the early stage of knee OA.

Vitamin D has been shown to influence the nociceptive pathway. Tague et al. found that vitamin D deficiency leads to hyperinnervation by nociceptors and hypersensitivity of mice skeletal muscles [34]. Lower levels of vitamin D are associated with higher mechanical sensitivity in patients with chronic pain [26]. Roesel hypothesized that a vitamin-D deficiency state might alter the functioning of descending inhibitory control pathways [35]. It also seems to have anti-inflammatory properties [36]. Amer and Quayyum found an inverse relationship between vitamin D and C-reactive protein in participants with a low level of vitamin D [37]. Similarly, Sun et al., 2021 showed that PTH attenuates OA pain by inhibiting subchondral sensory innervation, suggesting a critical role of PTH in OA pain [38]. Nevertheless, it is necessary to highlight that pain perception is a multifactorial experience involving the biological body state, cognitive and motivational-affective state, and the context of the individual [31]. Future studies should therefore interpret our results on vitamin D deficiency and PTH only as possible factors involved in the complexity of chronic pain.

In those patients with deficiency, we also found a statistically significant negative moderate relationship between the level of vitamin D and the level of disability. Vitamin D is essential for muscle health, as it plays a role in the maintenance of an adequate calcium plasma concentration and the differentiation of myoblast cells [35,36]. A deficiency might have altered physical performance during physical tests. Actually, Javadian et al. found a positive relationship between vitamin D concentration and quadriceps femoris muscle strength in patients with OA [23]. Future studies should evaluate the specific influence of vitamin D deficiency on the skeletal muscle properties of patients with EOA.

Finally, we also found a statistically significant positive moderate relationship between vitamin D concentration, PTH level, and social participation and a statistically significant negative strong to moderate relationship between vitamin D concentration and the presence of anxiety and depressive symptoms in that subgroup of patients. Sun exposure is the major source of vitamin D. Holick et al. warn that without sufficient sun exposure, it is not possible to produce the right amount of vitamin D [39]. Patients with higher anxiety levels, depressive symptoms, and level of pain intensity may therefore diminish their social participation and interaction, and consequently their sun exposure. Actually, Knippenberg et al. reported that there is a relationship between higher depressive symptoms and lower reports of sun exposure [40]. Future studies should assess the influence of psychological and social variables on vitamin D levels in patients with OA.

Our results on the relationship with social participation bring another factor into play, the level of physical activity [41]. It has been shown that most patients with early OA do not follow the recommendations from the Centers for Disease Control and Prevention and the American College of Sports Medicine for physical activity [42]. Higher increases of the self-reported or objective level of physical activity are associated with a higher increase of vitamin D, and participation in physical activity may increase the opportunity for sun exposure [34,43] Similarly, the practice of physical training has been shown to increase the concentration of PTH [35,36,37,38]. Future studies should focus on the interaction between the vitamin D concentration, PTH level, and the level of physical activity in patients with EOA.

### Limitations

Due to the nature of the study, it is not possible to assess any cause-and-effect relationship nor the temporal relationship between the deficiency of vitamin D, PTH level, and the different variables. It is not possible to determine the direction of the relationship (e.g., if the deficiency of vitamin D is the result of the presence of OA or a responsible for its incidence). We included only Caucasian participants from the Spanish population. Because skin color hugely influences the production of vitamin D, it is not possible to extrapolate results to the worldwide population [39].

## 5. Conclusions

Patients with knee EOA show a lower level of vitamin D than matched controls. The presence of vitamin D deficiency among patients with knee EOA is significantly related to a higher level of pain intensity, disability, anxiety and depressive symptoms, lower social participation, and physical performance. Lower PTH level is associated with higher pain intensity and lower social participation. These results highlight the relevance of vitamin D in the early diagnosis and prevention of EOA.

## Figures and Tables

**Figure 1 nutrients-13-04035-f001:**
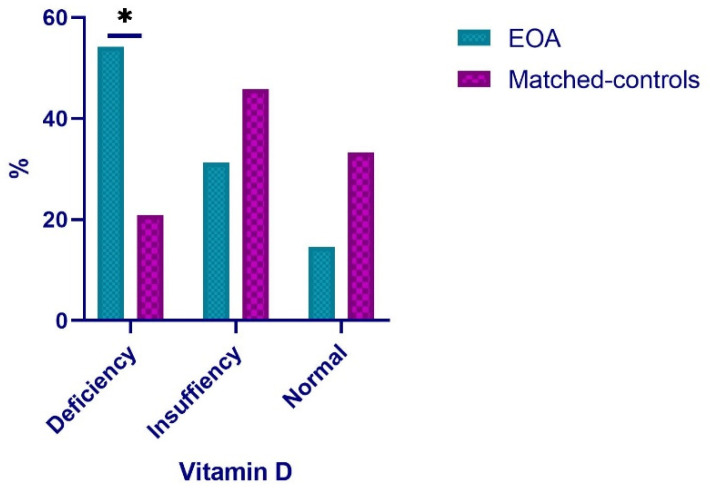
Distribution of EOA patients and matched controls according to vitamin D levels. * *p* < 0.05; EOA: Early Osteoarthritis.

**Figure 2 nutrients-13-04035-f002:**
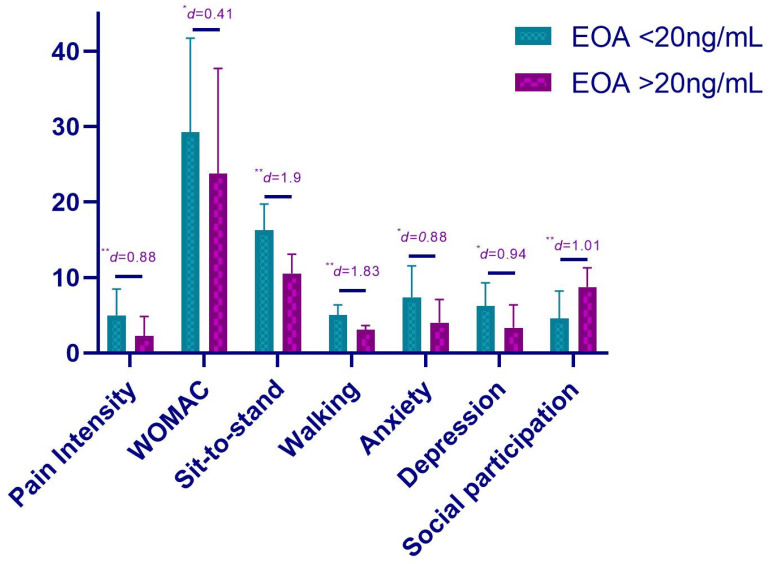
Physical and psychological status according to vitamin D levels. ** *p* < 0.01; * *p* < 0.05.

**Table 1 nutrients-13-04035-t001:** Descriptive and demographic variables.

Measures	EOA (*n* = 48)	Controls (*n* = 48)	*p* Value
Age (years)	52.36 ± 5.02	50.72 ± 6.87	0.46
BMI (kg/m^2^)	26.98 ± 4.36	27.50 ± 3.89	0.43
Gender			0.17
Women	39 (81.3)	43 (89.6)	
Men	9 (18.7)	5 (10.4)	
Economic status			0.86
Easy	8 (16.7)	3 (6.3)	
Fairly easy	35 (72.9)	38 (79.2)	
With some difficulties	3 (6.2)	7 (14.5)	
With great difficulties	2 (4.2)	0 (0)	
Alcohol			0.65
Never	10 (22.2)	8 (16.7)	
Seldom	20 (41.7)	18 (37.5)	
1–2 times/month	4 (8.3)	8 (16.7)	
1–2 times/week	7 (14.3)	12 (25)	
1 time day	5 (10.3)	2 (4.1)	
More than 1 a day	2 (4.1)	0 (0)	
Smoking			0.27
Yes	3 (6.25)	6 (12.5)	
No	10 (22.2)	20 (41.7)	
Ex	35 (72.9)	22 (45.8)	

Data are expressed as mean ± standard deviation or *n* (%). BMI: Body Mass Index; EOA: Early Osteoarthritis.

**Table 2 nutrients-13-04035-t002:** Vitamin D and PTH levels.

Measures	EOA	Controls	Mean Difference (95% CI)	Effect Size (*d*)
Vitamin D (Serum 25(OH)D concentration)	22.30 ± 7.3	29.31 ± 2.59	−7.01 ** (−10.39 to −1.23)	0.88
PTH concentration	40.78 ± 15.53	39.36 ± 14.76	−1.42 (3.15 to −7.69)	-

Data are expressed as mean ± standard deviation. EOA: Early Osteoarthritis; PTH: Parathyroid Hormone Levels. ** *p* < 0.01

**Table 3 nutrients-13-04035-t003:** Physical and psychological status in patients with EOA according to vitamin D levels.

Measures	EOA Deficiency (<20 ng/mL)	EOA Sufficiency (>20 ng/mL)	Mean Difference (95% CI)	Effect Size (*d*)
Pain intensity	4.95 ± 3.52	2.22 ± 2.59	2.73 ** (−3.39 to −0.76)	0.88
WOMAC Total	29.27 ± 12.45	23.76 ± 13.93	5.51 * (−14.65 to −6.64)	0.41
Sit-to-stand (s)	16.29 ± 3.44	10.53 ± 2.54	5.76 ** (−6.34 to −0.81)	1.90
Walking speed (km/h)	4.99 ± 1.38	3.07 ± 0.54	1.92 ** (−1.34 to −0.11)	1.83
Anxiety	7.36 ± 4.17	4.01 ± 3.05	4.35 * (−3.86 to −2.15)	0.88
Depression	6.21 ± 3.06	3.31 ± 3.06	2.9 * (−2.96 to −0.15)	0.94
Social participation	4.57 ± 3.61	8.73 ± 2.55	−4.16 ** (−2.82 to −1.14)	1.01

Data are expressed as mean ± standard deviation. ** *p* < 0.01; * *p* < 0.05.

**Table 4 nutrients-13-04035-t004:** Correlation analysis.

	Vitamin D (Serum 25(OH)D Concentration)	PTH
Pain intensity	−0.651 **	−0.686 **
WOMAC Total	−0.443 *	−0.234
Anxiety	−0.623 **	−0.153
Depression	−0.351 *	−0.041
Social participation	0.355 *	0.331 *

** *p* < 0.01; * *p* < 0.05. PTH: Parathyroid Hormone Levels.

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
