# Peer review of "The Role of Vitamin D in Early Knee Osteoarthritis and Its Relationship with Their Physical and Psychological Status"

_nutrients, 2021, doi:10.3390/nu13114035_

Round 1
Reviewer 1 Report
This is an appropriately conducted cross-sectional analysis reporting the association of serum Vit D and PTH in the early knee OA population with pain intensity, disability, psychological, and functional variables.
Major comment.
In the result section, the reporting of the primary outcome is somewhat hidden in the text and not coming out; I suggest the author appropriately present that as the introduction clearly point out that the primary aim of interest of this study is, “the aim of this cross-sectional study is to study the association of serum concentrations of vitamin D with knee EOA”
Hence the result section should be drafted in a way that the reader can easily find the outcome Re the primary aim of the study.
Minor suggestions.
- Please review the text below from the abstract, it needs clear writing.
“Osteoarthritis (OA) is a common joint condition one of the causes of disability worldwide”
- There are systematic reviews reporting the lack of effect of Vit D in the OA population. Please use the same also to present the introduction/background section.
“Rheumatol Int 37, 1489–1498 (2017). https://doi.org/10.1007/s00296-017-3719-0”
- Please describe controls were matched for what characteristics?
- Once Kellgren and Lawrence (KL) is used, please use abbreviation in the subsequent occurrence. Similarly, to be done of any abbreviations used in the manuscript.
- Line 298 says, “statistically significant negative moderate relationship between Vitamin D concentration and the presence of anxiety and depressive symptoms.”
The correlation of Vit D levels and anxiety (r=-0.623; p<0.01) was strong as mention by the author in the result section 3.3. Please make it uniform.
Reviewer 2 Report
The authors have studied vitamin D and PTH levels in patients with early osteoarthritis and tried to relate vitamin D and PTH levels with various health parameters in patients with early osteoarthritis.
It is well known that vitamin D and PTH levels are inverse related, as if vitamin D levels are low PTH levels increase. The authors should provide an explanation of their findings, namely the relationship of both low vitamin D and low PTH levels to various parameters of disease severity in patients with early osteoarthritis.
The use of the English language should be improved.
